# Vitamin D Deficiency is Associated with Handgrip Strength, Nutritional Status and T2DM in Community-Dwelling Older Mexican Women: A Cross-Sectional Study

**DOI:** 10.3390/nu13030736

**Published:** 2021-02-26

**Authors:** Luciano Mendoza-Garcés, María Consuelo Velázquez-Alva, María Fernanda Cabrer-Rosales, Isabel Arrieta-Cruz, Roger Gutiérrez-Juárez, María Esther Irigoyen-Camacho

**Affiliations:** 1National Institute of Geriatrics, Ministry of Health, Mexico City 10200, Mexico; iarrieta@inger.gob.mx; 2Health Care Department, Metropolitan Autonomous University, Unit Xochimilco, Mexico City 04960, Mexico; mcvelaz@correo.xoc.uam.mx (M.C.V.-A.); mcabrer@correo.xoc.uam.mx (M.F.C.-R.); 3Department of Biomedical Sciences, School of Medicine, Faculty of Higher Studies Zaragoza, National Autonomous University of Mexico, Mexico City 09230, Mexico; roger.gutierrez@zaragoza.unam.mx

**Keywords:** vitamin D deficiency, handgrip strength, muscle skeletal health, full mini nutritional assessment, aging, community dwelling

## Abstract

The aim of this study was to evaluate the association between handgrip strength, nutritional status and vitamin D deficiency in Mexican community-dwelling older women. A cross sectional study in women ≥ 60 years-old was performed. Plasma 25-hydroxyvitamin D (25(OH)D) concentrations were measured by a quantitative immunoassay technique. Handgrip strength was assessed using a dynamometer, while nutritional status was assessed through the Full Mini Nutritional Assessment (Full-MNA). A total of 116 women participated in the study, their mean age was 70.3 ± 5.8 years; 49.1% of the study group had plasma 25(OH)D levels lower than 40 nmol/L [16 ng/mL]. Meanwhile, 28.45% of participants had low handgrip strength (<16 kg), and 23.1% were identified at risk of malnutrition/malnourished according with Full-MNA score. Women with 25(OH)D deficiency (<40 nmol/L [16 ng/mL]) were more likely to have low handgrip strength (OR = 2.64, *p* = 0.025) compared with those with higher 25(OH)D values. Additionally, being malnourished or at risk of malnutrition (OR = 2.53, *p* = 0.045) or having type 2 diabetes mellitus (T2DM) (OR = 2.92, *p* = 0.044) was also associated with low 25(OH)D. The prevalence of low plasma 25(OH)D concentrations was high among Mexican active older women. Low handgrip strength, being at risk of malnutrition/malnourished, or diagnosed with T2DM was also associated with Vitamin D deficiency.

## 1. Introduction

Vitamin D is a steroid hormone that plays an important role in various physiological processes due to its pleiotropic effects on cells of several biological systems, such as the immune, cardiovascular, intestinal or skeletal muscle system. Relevant functions of vitamin D include synthesis of calcium-binding proteins and modulation of contractility in cardiomyocytes, proper absorption of calcium and phosphate ions in the intestinal epithelium as well as bone remodeling and mineralization [1,2,3]. These cellular functions are essential for preserving muscle strength and physical performance in adult life; interestingly, during the aging process these functions may decline dramatically leading to illness or disability in older adults. 

Vitamin D deficiency in the older adult population has been related to chronic diseases such as frailty, sarcopenia, osteoporosis, mild cognitive impairment, dementia or type 2 diabetes mellitus (T2DM) [4,5,6]. The public health significance of the strong relationship between decreased serum 25-hydroxyvitamin D (25(OH)D) level and older population is as a risk factor for the onset of various chronic diseases. The prevalence of vitamin D deficiency worldwide is high; for example, in the adult population of the USA, its prevalence, adjusted for potential predictors as obesity, age and poor physical activity, is 28.9% [7]. In the European Union, the prevalence vitamin D deficiency is variable across countries between 13% and 40%, considering age group, ethnic mix and latitude of the study population [8]. In Latin America, a deficiency of vitamin D has been reported in several countries [9]. In Mexico, for example, the prevalence of vitamin D deficiency was 62% in older adults influenced by factors such as gender, smoking habit, education and physical activity [10,11]. Growing scientific evidence has shown an important link between vitamin D deficiency and muscle health, especially in older adults. In this regard, a key problem is that the vitamin D cut-offs to determine deficiency or sufficiency of this hormone in older adults are not well defined and whether serum 25(OH)D level could be considered as a risk factor in the prognosis of chronic disease is not clear. 

The variability in the serum 25(OH)D levels over the years could be the result of several factors such as nutritional status, the presence of intestinal or chronic kidney disease, or even life style. In clinical practice, specific cut-offs limits have been proposed for its use in older adults: deficiency (<50 nmol/L (<20 ng/mL)); insufficiency (50–75 nmol/L (20–30 ng/mL)) and normal (>75 nmol/L (>30 ng/mL)). These vitamin D cut-off points are based on the serum parathyroid hormone levels and absorption of calcium from the gastrointestinal tract in the general population [12]. Additionally, different cutoff points have been associated to other conditions such as increased risk of fractures, falls, and myopathy in older people of Caucasian and Hispanic/Latino descent [4,11,13,14,15,16,17], as well as infectious and cardiovascular diseases [18,19]. In particular, in a longitudinal study based on data from the NHANES, a cutoff level of 40 nmol/L [16 ng/mL] was set because this plasma concentration complies with an intake equivalent to the Estimated Average Requirement [20]. 

Aging is associated with a gradual and progressive impairment of the musculoskeletal function and, when the loss of muscle mass and strength leads to falls, functional deterioration and disability, in older people, the condition is known as sarcopenia. This clinical entity has a very complex pathophysiology [21], the cellular and physiological mechanisms involved in muscle function are distinct for the muscle mass and muscle strength components. These differences are important to accurately define the concept of dynapenia vs the more widely known concept of sarcopenia. The term of dynapenia is defined as an age-related loss of muscle strength (maximal voluntary force) and mechanical power (product of force times velocity) [22,23]. 

A large portion of vitamin D requirements is obtained through sunlight exposure in young people; but the supply of this vitamin is reduced in the old age due to an inadequate nutrition and the presence of chronic diseases [24,25]. A poor nutritional status is a frequent health problem found in older adults, and in some populations, it has been found in association with sarcopenia and dynapenia [26,27,28]. Furthermore, it is well known that an inadequate diet is a determinant factor in vitamin D deficiency [29]. In this regard, an instrument used extensively by specialists to assess the nutritional status of older adults is the Mini Nutritional Assessment (MNA) score. This tool has been previously used to assess the association of vitamin D levels with nutritional status in older adults. Low MNA scores were associated to an increased likelihood of having decreased level of serum vitamin D in very old adults [30]; likewise, Santos et al., reported that the risk of malnourishment and malnutrition increased the risk of vitamin D deficiency among adults over 65 years old, from Nutrition UP 62 study [31]. Furthermore, there are studies that estimate intake rather than measure vitamin D plasma levels in the geriatric population [32,33]. Similarly, in clinical practice the vitamin D status of patients is not assessed systematically.

A number of studies carried out in clinical facilities have shown an association between dynapenia and vitamin D deficiency, suggesting a relevant role for vitamin D in muscle function in older adults. Specifically, there are reports of patients with hip fractures [34,35], and in vulnerable groups such as immigrants or people with disabilities [36,37]. In contrast with the above, information on older adults living in their communities is scarce [38,39,40].

On the other hand, a dramatic worldwide increase in the prevalence of T2DM has been observed in association with skeletal muscle disorders in older adults [41,42]. In this regard, epidemiological studies have shown a high prevalence of sarcopenia in older adults presenting poor nutritional status and T2DM [27,43,44]. 

Not enough epidemiological evidence is available in independent, community dwelling, older adults about the relationship between vitamin D levels, and muscle function, nutritional status and T2DM in Latin American populations, including Mexico. In this regard, it should be emphasized that ethnical background appears to play an important role in determining the circulating levels of vitamin D [20].

Thus, the main aim of the current study was to identify the association between handgrip strength, nutritional status, T2DM and vitamin D deficiency in Mexican, community-dwelling, older women.

## 2. Materials and Methods

### 2.1. Study Design 

A cross-sectional study was performed in a group of elder women who attended a social and sports center located in the southeast area of Mexico City. At this facility, people exercise (dancing, swimming, yoga and tai chi) and engage in social gatherings. The study was conducted in strict adherence to the Helsinki principles for medical research in human subjects. The research protocol was approved by the National Geriatric Institute in Mexico City (DI-PI-009/2018) and the research was conducted from February 2019 to February 2020.

The goals and procedures of the study were explained individually to each invited older adult. Those willing to participate signed a consent letter where the risks and benefits of the study were clearly explained. All the participants received counseling based upon the results of their evaluations. To be included in the study, the following criteria had to be fulfilled by the interested person: 60 years of age and older, and physically able to walk by themselves. Individuals with severe illnesses or disability as well as those unwilling to participate were excluded from the study. The information about the presence of preexisting chronic diseases, was obtained during the survey by asking the participant women an open-ended question about concurrent health conditions.

### 2.2. Sample Size

Sample size calculation was performed with the aim of testing whether the odds ratio, (OR), between two groups is different from the null value (Ho: OR = 1) [45]. A proportion of older adults with low plasma vitamin D level and low handgrip strength of 0.35, and a proportion of 0.10 in the group without vitamin D deficiency was assumed. An α = 0.05 and a power (1-β) of 0.80 was used. The calculated sample size was 98 individuals; then, assuming a non-response rate of 25%, a total of 123 individuals were sought to participate in the study. Of the total of 123 older women, five did not agree to take part in the study (response rate 95.9%). Additionally, pre–existing medical conditions were recorded in each one of the 118 older women, but two of them were excluded from the study since they had a critical illness or disability. In the end, a data from a total of 116 older women were included in the analysis. Socio-demographic information for each participant was collected through a questionnaire. 

### 2.3. Nutritional Status 

To assess the nutritional status of each participant, the 18-item version of the Mini-nutritional assessment (Full-MNA) tool, carrying a score ranging from 0–30, was used; after computing the scores, participant women were classified as malnourished (Full-MNA score <17.0), at risk of malnutrition (Full-MNA 17.0–23.5), and well-nourished (Full-MNA score >23.5) [46,47]. The anthropometric measurements were performed by a certified dietitian following standardized procedures [48,49,50]. Body mass index (BMI) was classified using the values suggested by the World Health Organization (WHO) [51]. Additionally, the Lipschitz criteria was applied in order to identify underweight women [52]. To assess the examiner’s reliability, 15 (13%) of the participants were evaluated in duplicate. The inter-examiner consistency of the anthropometric measurement results was 93%. 

### 2.4. Handgrip Strength 

For handgrip strength assessment, a mechanical hand dynamometer was used (TKK 5001; Takei Scientific Instruments, Tokyo, Japan), the measurements were performed in triplicate for each participant; the average of the two highest values was subsequently obtained [53]. Low handgrip strength (<16 Kg for women) was defined according to the value proposed in the revised algorithm for sarcopenia diagnosis from the European Working Group on Sarcopenia in Older People (EWGSOP) [54].

### 2.5. Activities of Daily Living

For the assessment of the Activities of Daily Living (ADL) we used the Barthel Index (BI) score, which takes values in the range of 0 to 100 [55]. Based on this score, the participants were classified as follows: “Totally dependent” (BI ≤ 20 points), “Severely dependent” (BI 21–60 points), “Moderately dependent” (BI 61–90 points), “Mildly dependent” (BI 91–99 points) or “Independent” (BI = 100 points).

### 2.6. Biochemical Assessment of 25(OH)D

Venous blood samples were drawn using a vacutainer blood collection system (BD Medical, New Jersey, USA), the blood cells were immediately separated by centrifugation, and the plasma was saved and stored at −20 °C.

25-hydroxyvitamin D was determined by a quantitative immunoassay technique using a commercial kit as (Abcam ELISA kit number ab213966, Cambridge, MA, USA) per the provider instructions; the assay had a sensitivity of 4.94 nmol/L [1.98 ng/mL]; the detection range was 1.25 nmol/L [0.5 ng/mL] to 2520.96 nmol/L [1010 ng/mL]. Colorimetric detection was performed on a Multiskan™ GO UV/Vis microplate spectrophotometer (Thermo Scientific, Waltham, MA, USA).

### 2.7. Statistical Analysis

Categorical variables are presented as percentages while numeric variables as means ± standard deviations. The Pearson Chi –square test was used for testing independence between categorical variables. Comparison of means was performed using linear regression models. 

To study the association between vitamin D status (cut off point 40 nmol/L [16 ng/mL]) and handgrip strength (cut off point = 16 kg), MNA (cut off point = 23.5), Barthel Index (cut off point = 100) and T2DM logistic regression models were constructed, and crude and age adjusted OR were obtained, including confidence intervals (95%CI). Interactions between nutritional status and handgrip strength were tested. To evaluate the goodness of fit of the models, we used the Hosmer-Lemeshow test. Statistical significance was set at α = 0.05. The statistical package STATA V 16 (College Station, TX, USA) was used for data analysis.

## 3. Results

In this study, 116 women over 65 years-old were included, the mean age was 70.3 (±5.8) year. Table 1 displays the demographic features of the study population, shows that less than a half of the participant women were married (44.8%), while about a third of them were widows (32.8%). Regarding the occurrence of chronic diseases, the most common condition was high blood pressure (HBP) (44.0 %), followed by T2DM which afflicted 17.2% of the study participants. The nutritional condition of the majority of the participants was normal (76.7%), but about 23.1% was either at risk of developing malnutrition or malnourished. The handgrip strength average value was 18.1 ± 4.0; 28.5% of participants in the study group had decreased handgrip strength (<16 kg). The results from the Barthel index assessments indicated that the majority of these older women were independent enough to carry out basic activities of daily living (68.1%) (Table 1). Bladder incontinence was the most common disorder found among the study participants accounting for 19.8% of all. Less than 5% of the participants had trouble walking, moving from their bed to a chair, climbing stair or bowel continence. Only one participant (0.9%) reported some degree of dependency for activities such as feeding, bathing, dressing, grooming, or toilet use. On the other hand, the participants had a mean plasma level of 25(OH)D was 58.4 ± 90.8 nmol/L (23.4 ± 36.38 ng/mL), however, values under 50 nmol/L (20 ng/mL) and 40 nmol/L (16 ng/mL) were observed in 60.3% and 49.1% of them, respectively. Only 16.4% of the participants had levels of 25(OH)D above 75 nmol/L (>30 ng/mL). No significant difference (p = 0.567) was found in the mean plasma concentrations of 25(OH)D between the group of 32 participants studied during the period Spring-Summer (mean 70.57 ± 134.9 nmol/L (28.27±54 ng/mL)) compared to the 84 participants studied during Fall-Winter (53.7 ± 67.5 nmol/L (21.52 ± 27 ng/mL)) period.

Table 2 shows the anthropometric characteristics of the study group. It shows that 30.2% of the study participants had a normal Body Mass Index (BMI) according to the WHO cut-offs points. However, using the Lipschitz criteria up to 46.1% of the participants were overweight, and only 10.4% were underweight.

The distribution of anthropometric variables, handgrip strength, Full-MNA and Barthel index with plasma level values of 25(OH)D above and below a cut-off point of 40 nmol/L (16 ng/mL) are shown in Table 3. None of the features shown was associated with the plasma levels of 25(OH)D (*p* = 0.393) except for two: handgrip strength (*p* = 0.014), and Full-MNA score (*p* = 0.002). No association between the mean score of basic activities of daily living (Barthel index) and 25(OH)D was observed (*p* = 0.498).

Table 4 shows the association between plasma levels of 25(OH)D and age, presence of T2DM, handgrip strength, nutritional status, and Barthel index score. These results showed that a high proportion of women with T2DM also have low levels of 25(OH)D compared to those who do not have the disease equally low levels of 25(OH)D (*p* = 0.040). Women with low handgrip strength (<16 Kg) showed lower levels of 25(OH)D more often than those with higher handgrip strength (*p* = 0.017). Importantly, women at risk of malnutrition or malnourished also had lower plasma 25(OH)D (*p* = 0.038). In contrast with the above results, Barthel index was not associated with 25(OH)D levels.

Along the same lines of data shown above, in Table 5, OR results from the logistic regression adjusted model indicated an association between T2DM and vitamin D deficiency in the participant women. In fact, women with the disease are 3 times as likely of having low levels of 25(OH)D than women without T2DM (OR = 2.92, *p* = 0.044). Similarly, older women with vitamin D deficiency were more than twice as likely of having low handgrip strength as those with higher vitamin D values (OR = 2.64, *p* = 0.025). Furthermore, the risk of malnutrition or frank malnutrition was associated to a higher probability of having lower levels of circulating 25(OH)D (OR = 2.53, *p* = 0.045), no significant association was found between vitamin D status and Barthel Index (*p* = 0.258). Lastly, no statistically significant interaction was observed between nutritional status and the decrease of handgrip strength as a function of vitamin D deficiency (*p* > 0.05).

## 4. Discussion

In this study an association between low circulating levels of vitamin D, under 40 nmol/L (16 ng/mL), and decreased handgrip strength was found in Mexican older women engaged in social and recreational activities. In agreement with our results, a U.S. study on a group of people 98 years of age and older, community dwelling or living in nursing homes, were studied found that plasma levels of 25(OH)D lower than 75 nmol/L (30 ng/mL) were associated with lower handgrip strength [56]. It is conceivable that the mechanisms underlying this association have to do with the known role of 25(OH)D or its metabolites in the regulation of genes and signaling pathways involved in calcium and phosphate homeostasis, as well as the proliferation and differentiation of skeletal muscle cells [57,58]. Vitamin D controls the initiation of muscular regeneration by promoting an increase of the cross-sectional area of skeletal muscle fibers through modulatory actions on the cell cycle of the skeletal tissue [59,60,61].

Numerous epidemiological studies have suggested a potential role for vitamin D in the maintenance of skeletal muscle function, physical performance, and the preservation of independence in older adults [62,63]. In this regard, several randomized clinical assays have examined the link between vitamin D supplementation and improved muscular strength. For example, a study on a cohort of 65 years-old patients with knee osteoarthritis, a significant improvement in grip strength and physical performance was observed with vitamin D supplementation, even though it did not improve knee extension [64]. Another study in Australia found that the strength of the hip muscles improved significantly after vitamin D_2_ supplementation with a 1000 IU/day dose during one year in older women with vitamin D deficiency [65]. Despite these findings, results from a meta-analysis in community dwelling people over 65 years-old, did not find any improvement of muscular strength after vitamin D and calcium supplementation [66].

The results showed that nearly one half of the population studied have low levels of vitamin D (<40 nmol/L (16 ng/mL)), in agreement with previous studies in Mexicans [10]. The mean plasma vitamin D value found in the current study was closely similar to that reported by the Mexican Health and Aging Study (2012) in women over 60-years old [11]. Interestingly, the vitamin D levels in this study were lower than those determined in samples of the population also over 60 years old but living in the southern states of Mexico [67]. It is possible that this is due to a higher sun exposure in communities in the south part of Mexico which are much less urbanized than Mexico City, and possibly also to differences of lifestyle. A study performed in the United States about the trends of vitamin D concentrations showed that non-Hispanic Whites but not Hispanics and Blacks-showed a slight trend towards an increase in vitamin D levels from the year 1998 to 2010; however, Mexican-Americans in this study did not show this variation [20]. Studies in Latin-American countries other than Mexico, have shown a deficiency of vitamin D in the adult population [9]. In a meta-analysis of data from various European countries reported various degrees of vitamin D deficiency [8]. These results suggest that deficiency of vitamin D is a prevalent problem in the older population of various regions of the world. Indeed, the oldest members of the population are at increased risk of developing vitamin D deficiency due to various causes including diminished skin synthesis secondary to limited sun exposure, chronic kidney disease (CKD) and malabsorption of the gastrointestinal tract [59,68]. Furthermore, there are reports of an aging associated decrease in the number of vitamin D receptors in skeletal muscle [69,70,71].

In the current study, we did not find seasonal differences in the levels of vitamin D throughout the year, which can be explained by the fact that Mexico City does not undergo major changes in the light/dark cycle as is the case in other world latitudes. This means that outdoors activity time in these populations is not subjected to marked variations. Several published studies have focused on geographic latitude as a modifying factor of vitamin D levels [72,73].

The results of the regression model in this study showed that women at risk of malnutrition or malnourished were more likely to have lower levels of vitamin D than were women with an adequate nutrition as assessed through the MNA. This nutrition evaluation tool not only has been validated and standardized in numerous human populations and allows to assess the nutritional status of older adults with various medical conditions [74]. Its full version, consisting of 18 items, includes four areas: anthropometric, general assessment, dietary assessment, and self-assessment [46].

In this regard, a study by Tsagari et al., demonstrated a correlation between MNA score and the plasma levels of vitamin D (rho = 0.685, *p* < 0.001), suggesting that this score could be used as a reliable surrogate marker for vitamin D levels [75]. In another study performed in a group of Spanish adults over 85 years old, the authors found that lower MNA scores were associated to a higher probability of having a low circulating concentration of vitamin D (<62.4 nmol/L (<25 ng/mL)) [30]. Similarly, in a group of Portuguese adults over 65 years old, was reported that participants classified either at risk of malnutrition or as malnourished based on the MNA score, also had an increased risk of vitamin D deficiency (< 30 nmol/L (<12 ng/mL)) when compared to participants in good nutritional condition [31]. When comparing these results with those of the current study, we have to bear in mind the heterogeneity of the various studies, especially regarding the cut off points for vitamin D levels, as well as the specific features of the subjects of study. Despite all this, the association between nutritional status (MNA score) and the deficiency of vitamin D appears to be consistent.

Dietary patterns could account for the association between nutritional status and vitamin D. For example, in a cross-sectional study the researchers observed that meat and fish consumers showed a higher level of vitamin D compared to vegetarian and vegan persons [76]. It has also been reported that among adults over 65 years-old high MNA scores are associated to the habit of consuming higher amounts of white meat, red meat and processed meat [77]. Despite the fact that daily intake in the diet is not the only determinant of the plasma levels vitamin D, a cross-sectional study by Nakamura et al. showed that the intake of meals rich in the vitamin (such as fish) may play a role in the maintenance of adequate vitamin D levels in the long term [78]. The current results suggest that the circulating levels of vitamin D should be taken into account in the assessment of the nutritional status of older adults, in light of the observation that patients with normal MNA scores may have low vitamin D levels, as reported in a population of Spanish older people by Formiga et al. [30] and in the current study. 

Our findings contribute to strengthen the concept that low levels of vitamin D are associated with the presence of T2DM. Furthermore, these results confirm this observation reported in only a handful of previous studies on similar cohorts of human subjects [79,80,81]. So far, the mechanisms underlying the association between vitamin D deficiency and T2DM have remained elusive due probably to the fact that most studies in the literature are descriptive in nature and have not focused on investigating the cause-effect relationships between vitamin D supplementation and the development of T2DM [82,83,84].

In vitro studies have shown that beta cells from human pancreatic islets of subjects with vitamin D deficiency are insulin resistant, thus directly linking the two phenomena [85]. Interestingly, it has been reported that rat pancreatic islets express both the mRNA and the protein of 1-alpha-hydroxylase (CYP27B1), an enzyme involved in the conversion of the inactive to the active form of vitamin D [86]. Additionally, in vivo studies have shown that vitamin D deficient rats displayed a phenotype characterized by decreased insulin secretion, a defect that is corrected after vitamin D supplementation [87]. Altogether, these studies provide clues about the possible mechanisms underlying the improvement of insulin sensitivity observed after supplementation with vitamin D, and may suggest a role of this defect in the pathophysiology of T2DM.

One limitation of the current study is that its cross-sectional design did not allow us to identify a cause-effect relationship. Furthermore, the number of participants in the study was small; notwithstanding the above, it was possible to establish associations regarding relevant aging markers such as handgrip strength and nutritional status.

## 5. Conclusions

This study confirms and extends the evidence indicating that vitamin D deficiency entails negative consequences on handgrip strength even in independent older women. This study found a relationship between a low circulating level of vitamin D and a decrease of handgrip strength, and higher risk of malnutrition. Similarly, women with T2DM had lower levels of vitamin D compared to their non-diabetic counterparts. 

## Figures and Tables

**Table 1 nutrients-13-00736-t001:** Sociodemographic characteristics, medical conditions, nutritional status, activities of daily living score, and plasma 25(OH)D concentration of community dwelling Mexican older women.

Characteristic	*n* = 116	
Age (years) mean ± sd	70.3 ± 5.8	
	n	(%)
60–64	12	(10.3)
65–74	79	(68.1)
≥75	25	(21.5)
Marital status		
Married	52	(44.8)
Widowed	38	(32.8)
Non-married / never married	26	(22.4)
Chronic diseases		
Hypertension	51	(44.0)
Diabetes	20	(17.2)
Cardiovascular disease	8	(6.9)
Osteoporosis	27	(23.3)
Osteoarthritis	16	(13.8)
Thyroid diseases	10	(8.6)
Depression	15	(12.9)
Dementia	1	(0.9)
Recent fracture	10	(8.6)
Autoimmune disease	8	(6.9)
Others	9	(7.8)
Nutritional Status		
Full-MNA (points), mean ± sd	25.4 ± 3.0	
Well-nourished	89	(76.7)
At risk of malnutrition	25	(21.6)
Malnourished	2	(1.7)
Handgrip strength		
Handgrip strength (kg), mean ± sd	18.1 ± 4.0	
<16 kg	33	(28.5)
≥16 kg	83	(71.5)
Activities of daily living		
Barthel Index (points), mean ± sd	97.9 ± 4.2	
Independent	82	(70.7)
Midly dependent	24	(20.7)
Moderately dependent	10	(8.6)
Severely dependent	0	(0.0)
Totally dependent	0	(0.0)
Vitamin D		
Plasma concentration 25(OH)D nmol/L, mean ± sd	58.4 ± 90.8	
Plasma concentration 25(OH)D ng/mL, mean ± sd	23.38 *±* 36.4	
25(OH)D < 40 nmol/L(<16 ng/mL)	57	(49.14)
25(OH)D ≥ 40 nmol/L(≥16 ng/mL)	59	(50.86)

**Table 2 nutrients-13-00736-t002:** Anthropometric characteristics of community dwelling Mexican older women.

Measurements	Total
	*n =* 116
Weight (kg) mean ± sd	61.6 ± 10.7
Height (cm) mean ± sd	150.9 ± 6.6
Body Mass Index (kg/m^2^) mean ± sd	27.1 ± 4.3
WHO classification	
Normal *n* (%)	35 (30.2)
Overweight *n* (%)	57 (49.1)
Obesity *n* (%)	24 (20.7)
Lipschitz classification	
Underweight *n* (%)	12 (10.4)
Normal *n* (%)	50 (43.5)
Overweight *n* (%)	53 (46.1)
Mid-upper arm circumference (cm) mean ± sd	30.0 ± 3.5
Calf circumference (cm) mean ± sd	34.5 ± 3.2

WHO, World Health Organization.

**Table 3 nutrients-13-00736-t003:** Anthropometric measurements, handgrip strength, MNA, and Barthel Index according to plasma 25(OH)D concentration in community dwelling Mexican older women.

Variable	Plasmatic Vitamin 25(OH)D Status		
	Vit D <40 nmol/L [16 ng/mL]	Vit D ≥ 40 nmol/L [16 ng/mL]	*p*
	*n* = 57	*n* = 59	
Age (years), mean ± SD	70.72 ± 6.51	69.80 ± 5.02	0.393
Weight (kg), mean ± SD	61.80 ± 11.78	61.49 ± 9.69	0.877
Height (cm), mean ± SD	150.63 ± 6.54	151. 08 ± 6.65	0.716
Body Mass Index (kg/m^2^), mean ± SD	27.17 ± 4.57	26.95 ± 4.07	0.787
Mid-upper-arm circumference (cm), mean ± SD	30.14 ± 3.95	29.84 ± 3.13	0.656
Calf circumference (cm), mean ± SD	34.19 ± 3.45	34.81 ± 3.01	0.304
Hand grip strength (kg), mean ± SD	17.19 ± 4.03	18.99 ± 3.75	0.014
Full-MNA (score), mean ± SD	24.51 ± 3.07	26.25 ± 2.71	0.002
Barthel Index (score), mean ± SD	97.45 ± 3.91	97.96 ± 4.16	0.498

**Table 4 nutrients-13-00736-t004:** Age, handgrip strength, MNA according to plasma 25(OH)D concentration in community dwelling Mexican older women.

Variable	Plasmatic Vitamin 25(OH)D Status		*Total* *n* *(Column %)*	*p*
	Vit D <40 nmol/L [16 ng/mL]	Vit D ≥40 nmol/L [16 ng/mL]		
	*n* = 57 *n (Row %)*	*n* = 59 *n (Row %)*		
Age				
≥70 years	25 (43.86)	32 (56.14)	57 (49.14)	0.264
<70 years	32 (54.24)	27 (45.76)	59 (50.86)	
T2DM				
Yes	14 (89.83)	6 (10.17)	20 (17.24)	0.040
No	43 (75.44)	53 (24.56)	96 (82.76)	
Hand grip strength				
<16 kg	22 (66.67)	11 (33.33)	33 (28.45)	0.017
≥16 kg	35 (42.17)	48 (57.83)	83 (71.55)	
Activities of daily living				
Mildly/moderately dependent	20 (58.82)	14 (41.18)	34 (29.31)	0.179
Independent	37 (45.12)	45 (54.88)	82 (70.69)	
Full - Mini Nutritional Assessment				
Risk/Malnourished	18 (66.67)	9 (33.33)	27 (23.28)	0.038
Normal	39 (43.82)	50 (56.18)	89 (76.72)	

**Table 5 nutrients-13-00736-t005:** Crude and adjusted logistic regression OR of handgrip strength and nutritional status, type 2 diabetes mellitus, and plasma 25(OH)D concentration in community dwelling Mexican older women.

Variable	OR Crude	OR 95% CI	*p*	OR Adjusted *	OR 95% CI	*p*
Hand grip strength	2.74	1.18–6.38	0.019	2.64	1.13–6.18	0.025
Full-MNA	2.56	1.04–6.33	0.041	2.53	1.02–6.26	0.045
Activities of daily living	1.74	0.77–3.91	0.181	1.61	0.70–3.69	0.258
T2DM	2.88	1.02–8.12	0.046	2.92	1.03–8.31	0.044

* OR adjusted by age. Cut off values of independent variables: age = 70 years, hand grip strength = 16 kg, Full-MNA score = 23.5, Barthel Index score = 100. 25(OH)D OR reference value 25(OH)D ≥ 40 nmol/L (16 ng/mL).

## Data Availability

The data presented in this study are available on request from the corresponding author. Some data are not publicly available due to participants’ confidentiality agreement.

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
