# Peer review of "Vitamin D Deficiency is Associated with Handgrip Strength, Nutritional Status and T2DM in Community-Dwelling Older Mexican Women: A Cross-Sectional Study"

_nutrients, 2021, doi:10.3390/nu13030736_

Round 1
Reviewer 1 Report
Table 1 should be reformatted to separate the continuous and the categorical variable reporting – this means authors would not need to continuously repeat “n (%)” throughout the table and could instead include it in the heading – this would make it much easy to read and digest this information.
Title should be edited to reflect the use of a categorical cut-off for vitamin D levels and not a continuous variable.
The rationale for this specific study is unclear, and it appears that several relevant studies are not considered in the introduction. Authors really need to explain what the “gap” is and how they are addressing it. Authors reference Gumeiero et al, with the statement “Interestingly, a few studies in humans 82 have shown an association between dynapenia and vitamin D deficiency, suggesting a relevant role for vitamin D in muscle function in older adults” – This study also used hand grip testing to correlate vitamin D status, so authors should be explicit as to what the differences are between this study and that one/what was lacking about that study. Haslam et al, who also demonstrated the correlation previously is only mentioned in the discussion, and again, what this study adds to that should be made clear. Authors should also consider Houston et al doi: 10.1093/gerona/62.4.440 who showed this correlation in 2009, Granlund et al, 10.1016/j.nutres.2018.07.009 and Dhanwal et al 10.1007/s11657-013-0158-8 who both demonstrated the correlation between hand grip strength and vitamin D levels in 2013, and Wang et al https://doi.org/10.1111/cen.13952 and Aspell et al doi.org/10.2147/CIA.S222143 who published on it in 2019. With this large body of evidence already demonstrating this correlation, authors need to be specific as to what this “new” study adds in terms of differences in methods, cohorts etc. There are also already several studies in the published literature on the relationship between hand grip strength measures and nutritional assessment tool scores.
It appears from the reference and scoring system that the authors are using an old (1996) 18 question version of the MNA (it was revised in 2009 to include fewer questions and to have a lower total score). They authors should be explicit about this in the methods section see https://www.mna-elderly.com/ and https://www.mna-elderly.com/mna_forms.html which has details about the preferred terminology of “full MNA” referring to the 18-item MNA and “MNA” referring to the new short form. Keeping the terminology consistent and explicit will help clinicians and other researchers to more easily understand your work.
The introduction refers to <50nmol/L as the threshold for deficiency, but the analysis uses a threshold of <40nmol/L – this should be explained/justified. Also note there is a typographical error in the methods which
The results for the relationship between vitamin D plasma level category and MNA score are presented using MNA as a continuous variable – with means of 24.51 and 26.25 presented – although these may be “significantly” different they are both well above the threshold for well nourished (>23.5). Researchers should consider and comment on the usefulness of the continuous variable analysis and should consider analysing this a categorical variable to give it meaning. Researchers should also comment on the fact that the majority of the population did score in the well nourished category and discuss the potential issues with this in interpreting the results presented. Additionally, the use of ”nourishment”/MNA in the study needs to be better justified and as, as the authors say themselves “it is well known that 91 an inadequate diet is a determinant factor in vitamin D deficiency [29].”
Sensitivity and specificity for the vitamin D assay are reported in ng/mL data is reported in nmo/L – units should be converted and kept consistent.
The % values reported for the vitamin D categories in Table 1 to not add up correctly 97 people is not 83.62% of 116 people.
The sample size was reportedly calculated for a proportion of 0.10 of the group not having vitamin D deficiency. A) Why? B) this does not reflect the population collected and so it should be considered how this impacts the power. Similarly the proportion with low hand grip strength used in the calculation does not match the hypothesised value.
Sample size calculation was performed with the aim of testing whether the odds ra- 119 tio, (OR), between two groups is different from the null value (Ho: OR = 1) [31]. Data about 120 the association between vitamin D deficiency and handgrip strength was used for this 121 calculation [23]. A proportion of older adults with low plasma vitamin D level and low 122 handgrip strength of 0.35, and a proportion of 0.10 in the group without vitamin D defi- 123 ciency was assumed. An α= 0.05 and a power (1- β) of 0.80 was used. The calculated sam- 124 ple size was 98 individuals; then, assuming a non-response rate of 25%, a total of 123 in- 125 dividuals were sought to participate in the study. Of the total of 123 older women, 5 did 126 not agree to take part in the study (response rate 95.9%). Additionally, pre–existing med- 127 ical conditions were recorded in each one of the 118 older women, but 2 of them were 128 excluded from the study since they had a critical illness or disability. In the end, a data 129 from a total of 116 older women were included in the analysis. Socio-demographic infor- 130 mation for each participant was collected through a questionnaire
(MNA) tool was used; after computing the scores, participant women were classified as 134 malnourished (MNA score < 17.0), at risk of malnutrition (MNA 17.0 – 23.5), and well- 135 nourished (MNA score > 23.5)
There are lots of data presented eg medical conditions, activities of daily living score (Barthel), antropometry etc that do not have the origins this data, nor the rationale for including it, described in the methods or introduction.
Reviewer 2 Report
The paper from Mendoza-Garcés is interesting and results, even if not new, are well presented and the discussion is appropriate. I have some minor points:
- Why did you choose the cut-of of 40 nmol/l instead of those reported in the literature and suggested by guidelines?
- I would report in brackets also 25OHD values in ng/ml
- Did you test your method of measurement (immunoassay) by DEQAS international standardization programme?
Round 2
Reviewer 1 Report
I initially had LOTS of concerns about this paper, but the authors have made significant efforts to edit to address these queries.
In my initial review I asked what this study added to the body of knowledge as there have been many studies on the relationship between grip strength and vitamin D levels already. The authors have responded by adding the following to the text “Not enough epidemiological evidence is available in active, community dwelling older adults regarding the relationship between vitamin D levels, and muscle function and nutritional status in Latin American populations including Mexico.” And have indicated in their point-by-point response that this study focuses on ACTIVE women – however more than 30% of the population had a Barthel score of Dependent/Needs assistance – so this in incongruent with defining this population as “active” – authors should explain and consider additional analysis to address this (eg. stratified analysis by Barthel score) or multivariable modelling.
I suggest adding the specific population to the title
Why is barthel score not in table 5?
What are the adjustment applied in table 5? The statistics section only describes the statistics for the primary association of hand grip strength and vitamin D. It does not describe all the other analysis presented. This is very important to make clear.
“Similarly, older women with low handgrip strength were more than twice as likely of having vitamin D deficiency as those with higher handgrip values (OR =2.82, p=0.022).” – this appears to be the wrong way around and implying that hand grip strength impacts vitamin D levels, when the hypothesis is surely that vitamin D levels impact hand grip strength? Please review these analysis and the phrasing to ensure the correct directionality for the dependent and independent variables.
I am still unclear on what the MNA data adds to this manuscript or the body of knowledge here. The entire discussion paragraph on MNA is about how the results is known and expected. Despite the edits and response I still cannot see why this is presented or included in this study. It feels like “padding”. Furthermore, the T2DM results don’t seem to fit with the rationale or the title either the discussion says “Our findings contribute to strengthen the concept that low levels of vitamin D are associated with the presence of T2DM.” but the title of the paper is “Vitamin D Deficiency is Associated with Low Handgrip Strength and Nutritional Status in Community-dwelling Older Women: A Cross- Sectional Study” No mention of diabetes! Other than this condition also being associated with Vit D deficiency (which is already known as stated in the introduction) how does this relate to the research question? What is the rationale.
Please clarify if the pre-existing medical conditions data was self reported and how it was collected? Was there a list or was it open answer?
Please make sure what you are discussing and presenting matches to your justification, title and research question and that you explain it clearly. There seems to be lots of data just presented because you have it. I am fine with you characterising you cohort clearly, but please be clear on the primary aims and results and what is just an FYI.
